# Determinants of podoconiosis among residents of Machakle District East Gojjam Zone Amhara Region Ethiopia

**Teshome Tefera, Kassawmar Angaw Bogale, Yiteka Tegegn, Abebaw Gedef Azene, Kebadnew Mulatu, Gizachew Tadesse Wassie**◉*

Department of Epidemiology and Biostatistics, School of Public Health, College of Medicine and Health Sciences, Bahir Dar University, Bahir Dar, Ethiopia

* leulgzat@gmail.com

**Data Availability Statement:** The datasets used and/or analysed during the current study are available within the manuscript.

## Abstract

### Background

Podoconiosis (endemic non-filarial elephantiasis) is a chronic disease characterized by the development of persistent swelling of plantar foot initially; which progresses to the dorsal foot and lower leg slowly or in a number of acute episodes to reach the knee. About 4 million people are said to be affected by the disease worldwide and it is deemed a serious public health problem in at least 10 African countries including Ethiopia. Therefore this study aimed to identify the determinants of podoconiosis among residence in Machakel district.

### Method

Unmatched case control study design was conducted at Machakel district from August 30 to September 30, 2022. The sample size calculated using Epi-info software yielded 211 controls and 106 cases (317 study participants). Simple random sampling technique was used to select the cases using registration books of the district. Data were entered to Epi info version 7 and exported to SPSS version 22 for statistical analysis. Binary logistic regression was used to identify explanatory variables.

### Result

A total of 312 study participants (104 cases and 208 controls) were included giving a response rate of 98.42%. Bare foot (AOR, 5.83 [95% CI: 2.34–14.50]), female sex (AOR, 4.25 [95% CI: 2.22–8.14]), family history of podoconiosis (AOR: 3.01(95% CI: 1.41–6.42) and age group 41–60 (AOR: 5.05(95% CI: 2.35–10.83), and 61–80 AOR 15.74 95% CI: (5.56–44.55) were determinants of Podoconiosis.

### Conclusion and recommendation

Barefoot, sex, family history of podoconiosis and age were determinants of Podoconiosis. District health office should encourage at risk populations especially older people and

**Funding:** The author(s) received no specific funding for this work.

**Competing interests:** The authors declare that they have no competing interests.

individuals with family history of podoconiosis about shoe wearing practice all the time and not to expose their skin and feet.

## Author summary

Podoconiosis is a chronic neglected tropical disease that causes swelling in the feet and legs. It can lead to lifelong disabilities and is preventable. This research aimed to identify the factors that contribute to the development of podoconiosis among people in Machakle District East Gojjam Zone Amhara Region Ethiopia. The study found that regularly walking barefoot, having a family history of podoconiosis, being female, and being older were all risk factors for the disease. The findings suggest that promoting the use of shoes, especially among females, should be a priority in healthcare efforts to eliminate podoconiosis in Ethiopia and other similar settings.

## Introduction

Podoconiosis is a chronic disease characterized by the development of persistent swelling of plantar foot initially; which progresses to the dorsal foot and lower leg slowly or in a number of acute episodes to reach the knee. Finally, the disease may end up in a permanent feature of elephantiasis of varying degree. The disease is common in families of barefooted agriculturalists of tropical Africa [1].

Podoconiosis (from the Greek word for foot: podos, and dust: konos) is unique in being an entirely preventable non-communicable tropical disease, usually crystalline blockage of the limb lymphatic, almost always affecting the lower limbs, especially the feet. In local communities, it is often called 'mossy foot disease', because the skin becomes rough and bumpy and its appearance resembles moss [2].

Podoconiosis has a curable pre-elephantiasis phase. However, once elephantiasis is established, podoconiosis persists and may cause lifelong disability[3]. Podoconiosis (endemic non-filarial elephantiasis) has been recognized as a specific disease entity for over one thousand years and it is widespread in tropical Africa, Central America and north India, Indonesia, Colombia, Ecuador, Brazil and Sri Lanka yet, it remains a neglected and under-researched condition [4,5].

Podoconiosis (endemic non-filarial elephantiasis) is a geochemical disease occurring in individuals exposed to red clay soil of volcanic origin [6]. The disease causes bilateral, but asymmetrical swelling almost invariably of the lower legs [7]. Early symptoms of podoconiosis include itching of the skin of the forefoot and recurrent episodes of burning and oedema of the foot or lower leg, especially after periods of intense physical activity [8].

Although the aetiology is not fully understood, existing scientific evidence suggests the important role of exposure to irritant red clay soil in endemic areas as well as the effect of genetic susceptibility [9]. Podoconiosis is found in highland areas of tropical Africa, Central America and north-west India. Areas of high prevalence have been documented in Uganda [10], Tanzania [11], Kenya [12], Rwanda, Burundi, Sudan and Ethiopia [13], and in Equatorial Guinea [14], Cameroon [15], the islands of Bioko, Sao Tome & Principe [16] and the Cape Verde islands. And it is related to poverty. Studies have also indicated that podoconiosis exists in areas where the altitude is above 1000meters above sea level and annual rainfall above 1000 millimetres. About 4 million people are said to be affected by the disease worldwide and it is deemed a serious public health problem in at least 10 African countries [1,17,18,19].

Podoconiosis follows a chronic course causing progressively increasing disability with continued exposure to irritant soils. It results in bilateral progressive chronic swelling of the lower legs, usually limited below the level of the knees. The pathogenesis of the disease has not yet been investigated in depth, but it is believed to be caused by fine particles in the soil that penetrate the skin and induce an inflammatory reaction in the lymphatic system [20]. However, early stage disease can easily be treated by foot hygiene, bandaging and shoes.

Podoconiosis is classified into five stages where the first stage swelling is limited to below the ankle and is reversible overnight. The second stage swelling is not reversible, and when bumps and knobs are present they remain below the level of the ankle. In the third stage of the disease, bumps and knobs are found above the level of the ankle. The fourth stage entails above knee swelling whereas the fifth stage involves joint fixation as a result of surrounding soft tissue overgrowth [20].

Podoconiosis has recently been included in the World Health Organization's Neglected Tropical Diseases (NTDs) list [21]. Areas of high prevalence of podoconiosis have been documented in tropical Africa, Central America and north India [22]. Of affected countries, Ethiopia appears to have the highest number of people with podoconiosis, with 11 million people at risk through exposure to irritant soil, and an estimated 1 million people affected countrywide [23,24]. In Ethiopia, prevalence estimates range from 2.8 to 7.4% in endemic areas [23–26]. Podoconiosis can be prevented, early forms of the disease can be treated, and disease progression can be controlled with simple but effective measures such as washing feet with soap and water on a regular basis and wearing protective shoes consistently [1]. Hence this study aimed to study the determinants of podoconiosis in Machakel district Ethiopia.

## Methods and materials

### Ethics approval and consent to participate

The ethical clearance issues of this study were reviewed and approved by the Ethical Review Committee (IRB) of the College of Medicine and Health Sciences, Bahir Dar University with the ethical clearance reference number of 523/2022. Permission letter was also obtained from Amhara Public Health Institute (APHI) before the actual data collection; permission was taken from East Gojjam zone health department and Machakel woreda health office. Written informed consent was taken for each participant. Confidentiality was kept and their name was changed to codes.

### Study area

This study was conducted in Machakel district. Machakel district is one of the districts in East Gojjam zone, Amhara national Regional State. It is found at a distance of 328 km North West from Addis Ababa, the capital city of Ethiopia, 237 km far from Bahir Dar, the capital city of Amhara Region. Machakel woreda is bordered on the North by Bibugn district, on the South by Debre Elias district, on the Northwest by Sinan, on the Southwest by Gozamen and on the East by Dembecha (West Gojjam).According to the 2022 projected census 2007, the total population is 146,942 from which about 73,618(50.1%) are female and 73,324(49.9%) male population. Machakel district has 30 kebeles. In the Machakel district, there are 24 health posts and six health centres all are providing health service.

### Study design and period

Unmatched case control study design was conducted from August 30 to September 30, 2022.

## Source and study population

All resident of Machakel district were the source population. Whereas the study population was all adult patients age 18 years and older that were identified and registered as podoconiosis case by the district and neighbouring individuals without podoconiosis.

**Cases:**  Were person of age 18 years and older, resided in any kebeles of Machakel district and who had been diagnosed and registered as podoconiosis case by the district.

**Controls:**  Were persons age 18 years and older who did not have podoconiosis after clinically diagnosed by clinical nurses and who lives in the neighbouring house to podoconiosis patients (case).

## Sample size determination

The sample size was calculated using Epi-info 7 software based on the assumption of 95% confidence interval, 85% power, control to case ratio of 2:1, presence of exposed family history among controls 11.4%, odds ratio to be detected as 2.81[15] and non-response rate of 10% yielding 211 controls and 106 cases (317 study participants).

## Sampling techniques

Simple random sampling technique was used to select the cases from podoconiosis registration books of Machakel woreda health office as a sampling frame and control groups from the neighbouring of the cases.

## Variables

Dependent variable
Podoconiosis (yes/no)

Independent variables: The following variables were identified from different previous studies and assessed in the current study.

**Socio-demographic Characteristics** such as: Age, Sex, Residence, Income, Educational status, and Occupation.

**Behavioural related factors:** Feet washing and Shoe Wearing practices.

**Family history of podoconiosis:** It was whether the participants had podoconiosis case from either of 1rst degree, 2nd degree, 3rd degree or others family members or not.

## Operational definitions

**Feet washing:** It is the status of patients' daily wash practice of their feet with soap.

**Have enough water for washing**: It is the participants' perceived response whether water is always available for washing or not.

**Barefoot:** It is a status of participants not wearing any type of shoe every day.

**Family history:** History of podoconiosis in the family clustering, such as,1st degree (parents, child), 2nd degree (grandparents, siblings), 3rd degree (aunt, uncle, nephew, cousin, niece), other (husband and wife) [11].

**Shoe wearing:** wearing a full covering shoe every day during each activity.

## Data collection tools and techniques

Data were collected using interviewer administered questionnaire prepared in English language and translated into Amharic language for the purpose of community level and back to English language for checking consistency. Data was collected by clinical nurses.

## Data quality control

Questionnaire were prepared in English version by reviewing different literatures and translated in to Amharic (local language) and back to English by different language experts for consistency. Steps were taken to ensure the quality of this work. Pre-test was conducted on 16(5%) study participants in another district and necessary corrections were made on the questionnaire. The supervisors and principal investigator closely followed the day to day data collection process to ensure completeness and consistency of the collected questionnaires on a daily basis. Training was given for the data collectors and supervisors, and the whole data collection process was closely supervised.

## Data processing and analysis

Prior to analysis, the whole data was cleaned and checked for completeness. Errors related to inconsistency were verified using cross tabulation and other data exploration methods. The data was entered into Epi-Info version 7.2.1.0 software packages then transferred to SPSS version 22 software for analysis. Descriptive statistics was used to give a clear picture of background variables like age, sex, and other variables. The frequency distribution of both dependent and independent variables were done. Binary logistic regression was used to identify the determinants of podoconiosis with the outcome variable. Hosmer-Lemeshow test was used to check goodness of fit of the model. Variables having an association with the outcome variable a p-value of less than 0.2 were considered in the multivariable logistic regression analysis. Adjusted odds ratios (AORs) with 95% confidence intervals were used to show association between explanatory variables and a dependent variable. Those independent variables with P-value $< 0.05$ was considered statistically significant factors associated with outcome variable.

## Results

### Socio-demographic characteristics of participants

A total of 312 participants (104 cases and 208 controls) were included, representing a response rate of 98.42%. The mean age of respondents was 47.21 years (SD = 11.47 years). Nearly two third, 229(73.32%) were farmers by occupation. The mean age of the cases was 53.8 years (SD = 15.43 years). Female constitute 67.30% of cases, 58(55.76%) of cases had family history of podoconiosis, the proportion of bare foot among cases was 30.76% and 76.92% of cases cannot read and write. The mean age of controls was 43.88 years (SD = 9.5 years). Having family history of podoconiosis was recorded in (57.69%), bare foot (11.53%), female (44.71%) and being a farmer occupation constitutes (71.15%) of controls (Table 1).

### Behavioural related factors of participants

The proportion of bare foot in cases was 32(30.76%) whereas it was 24(11.53%) among controls. About 84(80.76%) of cases and 188(90.38%) of controls did wash their feet 6–8 times per week without soap and 52(50%) of cases, and 152(73.07%) of controls did wash their feet 6–8 times per week with soap.

Among cases, 26(25%) and 9(8.6%) did wear shoes at work and at home respectively. Whereas 88(42.30%) and 38(18.26%) wear shoes at work and at home respectively among controls. About twelve percent 12 (11.5%) of cases and 120(57.69%) of controls started shoe wearing between one to 20 years of age. More than half 58(55.76%) of cases and 120(57.69%) controls had a family history of podoconiosis (Table 2).

**Table 1. Socio-demographic characteristics of study participants in Machakel district North-West Ethiopia, 2022.**

| Variables | Category | Cases (n (%)) | Controls (n (%)) |
|---|---|---|---|
| Sex | Male | 34(54.8) | 115(60) |
| | Female | 70(45.19) | 93(44.71) |
| Age | 21–40 | 14(13.46) | 93(44.71) |
| | 41–60 | 63(60.57) | 101(48.55) |
| | 61–80 | 27(25.96) | 14(6.73) |
| Residence | Urban | 16(15.38) | 29(13.9) |
| | Rural | 88(84.6) | 179(86.05) |
| Marital status | Unmarried+ | 32(30.7) | 13(6.25) |
| | Married | 72(69.23) | 195(93.75) |
| Occupation | Merchant | 4(3.8) | 11(5.25) |
| | Farmer | 81(77.88) | 148(71.15) |
| | House wife | 16(15.38) | 47(22.59) |
| Education | Cannot read and write | 80(76.92) | 112(53.84) |
| | Can read and write | 21(20.19) | 67(32.21) |
| | Primary | 2(1.92) | 28(13.65) |
| | Higher education | 1(0.96) | 0(0) |
| Income(Ethiopian Birr) | 500–1500 | 48(46.15) | 48(23.41) |
| | 1501–3500 | 43(41.34) | 118(57.56) |
| | >3500 | 13(12.5) | 42(20.2) |

## Factors associated with Podoconiosis

Being barefooted, sex, family history of podoconiosis, occupation, marital status, age and ownership of a pair of shoes were associated with Podoconiosis in the uni-variable binary logistic regression analysis at p values< 0.25.

In multivariable logistic regression analysis: sex, being barefooted, family history of podoconiosis and age were associated with Podoconiosis with 95% CI, at a p-value <0.05 statistical significance level. The goodness-of-fit statistics for the model were assessed using the Hosmer-Lemeshow test; and the p-value was 0.97.

The odds of having Podoconiosis were 5.83 times higher among barefooted persons (AOR 5.83, 95% CI: 2.34–14.5) than whose who wore shoes. The odds of having Podoconiosis were 4.25 times higher in females (AOR 4.25: 95% CI: 2.22–8.14) than in males. Besides, the odds of having Podoconiosis were 3.01 times higher among participants those who had a family history of podoconiosis (AOR = 3.01, 95% CI: 1.41–6.42) compared with who had no a family history of podoconiosis. Furthermore, the odds of having podoconiosis were 2.05 times higher in age group 41-60(AOR = 5.05, 95% CI: 2.35–10.83) and 15.74 times higher in age group 61–80 (AOR = 15.74, 95% CI: 5.56–44.55) as compared to age less than 40 years (Table 3).

## Discussion

Even though podoconiosis has been known for more than a millennium, it has been a neglected and under-researched. Podoconiosis has recently been designated neglected tropical disease status by the WHO [25].

Barefooted people were at higher risk of developing podoconiosis as compared to those who wear shoes. This finding agreed with the study conducted in southern Ethiopia [26]. This implies barefooted individuals are more likely to be exposed to irritant minerals found in earth that lead to Podoconiosis [27,28].

**Table 2. Behavioural related factors of podoconiosis in Machakel district North-West Ethiopia, 2022.**

| Characteristics | Category | Cases (n (%)) | Controls (n (%)) |
|---|---|---|---|
| Having enough water for washing | No<br>Yes | 104(100) | 208(100) |
| Washing feet without soap per week | 1–2 times | 2(1.92) | 0(0) |
| | 3–5 times | 17(16.34) | 13(6.25) |
| | 6–8 times | 84(80.76) | 188(90.38) |
| | >8 | 1(0.96) | 7(3.36) |
| Washing feet per week with soap | 1–2 times | 5(4.80) | 10(4.80) |
| | 3–5 times | 33(31.73) | 35(16.82) |
| | 6–8 times | 52(50) | 152(73.07) |
| | >8 | 0(0) | 2(0.96) |
| Barefooted habit | Yes | 32(30.76) | 24(11.53) |
| | No | 72(69.23) | 184(88.46) |
| Starting year of shoe wearing | 1–20 year | 12(11.53) | 120(57.69) |
| | 21–40 year | 53(50.96) | 59(28.36) |
| | 41–60 | 7(6.73) | 5(2.40) |
| Years with shoe wearing since starting | All years | 30(28.84) | 181(87.01) |
| | More than half of the years | 22(21.15) | 2(0.96) |
| | Less than Half of the years | 20(20.00) | 1(0.48) |
| Number of days in a week wearing shoe | Every day | 47(45.19) | 194(93.26) |
| | More than 5 day | 22(21.15) | 4(1.95) |
| | 2–5 day | 35(33.65) | 10(4.80) |
| Where did you wear your shoes | At work place<br>At home | 26(25)<br>9(8.6) | 88(42.30)<br>38(18.26) |

**Table 3. Factors associated with podoconiosis in Machakel district North-West Ethiopia, 2022.**

| Variables | Category | case | Control | COR [95% CI] | AOR [95% CI] |
|---|---|---|---|---|---|
| Bare foot | No | 72 | 184 | 1 | 1 |
| | Yes | 32 | 24 | 3.40(1.87–6.17) | 5.83(2.34–14.50)* |
| Sex | Female | 70 | 93 | 2.54(1.55–4.16) | 4.25(2.22–8.14)* |
| | Male | 34 | 115 | 1 | 1 |
| Marital status | Single | 3 | 6 | 1 | 1 |
| | Married | 80 | 195 | 0.82(0.2–3.36) | 0.19(0.03–1.11) |
| | Divorced | 11 | 3 | 7.33(1.11–48.26) | 0.83(0.09–7.5) |
| | Widowed | 10 | 4 | 5.00(0.82–30.46) | 0.34(0.04–2.94 |
| Age group | 21–40 | 14 | 93 | 1 | 1 |
| | 41–60 | 63 | 101 | 4.14(2.17–7.89) | 5.05(2.35–10.83)* |
| | 61–80 | 27 | 14 | 12.81(5.44–30.14) | 15.74(5.56–44.55)* |
| Family history | No | 46 | 88 | 1 | 1 |
| | Yes | 58 | 120 | 1.09(1.21–3.11) | 3.01(1.41–6.42)* |
| Occupation | Farmer | 81 | 148 | 0.36(0.06–2.22) | 0.62(0.07–5.35) |
| | House wife | 16 | 47 | 0.22(0.03–1.48) | 0.27(0.02–2.74) |
| | Merchant | 7 | 13 | 1 | 1 |
| Owned a pair of shoe | No | 4 | 3 | 2.73(0.6–12.44) | 2.5(0.38–16.53) |
| | Yes | 100 | 205 | 1 | 1 |

1 = Reference

* = significant at p-value <0.05 in multivariable logistic regression

Being female was another variable with higher odds among cases compared to males. This finding was in line with a study done in the West and East Gojjam zones [15]. This might be due to exposure intensity variation. Traditionally, males usually wear shoes more frequently than females in this community.

Family history of podoconiosis was also at higher odds among cases than controls. This might be due to the effect of family genes in the development of Podoconiosis [29]. This study was in line with a study done in West and East Gojjam zones of Ethiopia [16].

Older people, particularly those aged 41–60 years and 61–80 years were at higher risks of developing podoconiosis as compared to those with an age group less than 40 years. This implies older individuals might have long-term exposure to irritant minerals on the earth that could increase the risk of developing podoconiosis.

## Limitation of the study

The limitation of this study was the possibility of recall bias that might have been introduced due to retrospective nature of the study design.

## Conclusion

Podoconiosis is a common but neglected tropical disease, leading to dramatic non-filarial elephantiasis in the tropics region. This study showed that the determinants of Podoconiosis are regular walking barefooted, family history of podoconiosis, female sex and older age. People should be encouraged to wear shoe all the time. Particularly, females in the study area and other similar settings should be given attention in the existing primary health care packages to promote shoes wearing.

## Acknowledgments

We would like to thank Machakel District Health Office workers for giving us baseline data and supportive letter for data collection. Also, we would thank the data collectors and the study participants.

## Author Contributions

**Conceptualization:** Teshome Tefera.

**Data curation:** Teshome Tefera.

**Formal analysis:** Teshome Tefera.

**Funding acquisition:** Teshome Tefera.

**Investigation:** Teshome Tefera.

**Methodology:** Teshome Tefera, Kassawmar Angaw Bogale, Yiteka Tegegn, Abebaw Gedef Azene, Kebadnew Mulatu, Gizachew Tadesse Wassie.

**Resources:** Teshome Tefera.

**Software:** Teshome Tefera, Gizachew Tadesse Wassie.

**Supervision:** Gizachew Tadesse Wassie.

**Validation:** Gizachew Tadesse Wassie.

**Visualization:** Gizachew Tadesse Wassie.

**Writing – original draft:** Teshome Tefera, Gizachew Tadesse Wassie.

**Writing – review & editing:** Teshome Tefera, Kassawmar Angaw Bogale, Yiteka Tegegn, Abebaw Gedef Azene, Kebadnew Mulatu, Gizachew Tadesse Wassie.

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
