## [Decision Letter · Decision Letter 0]

13 Jun 2023

Dear Mr. Wassie,

Thank you very much for submitting your manuscript "Determinantes of Podoconiosis among Residents in Machakle District East Gojjam Zone Amhara Region Ethiopia" for consideration at PLOS Neglected Tropical Diseases. As with all papers reviewed by the journal, your manuscript was reviewed by members of the editorial board and by several independent reviewers. In light of the reviews (below this email), we would like to invite the resubmission of a significantly-revised version that takes into account the reviewers' comments. 

We cannot make any decision about publication until we have seen the revised manuscript and your response to the reviewers' comments. Your revised manuscript is also likely to be sent to reviewers for further evaluation.

Sincerely,

Victor S. Santos, Ph.D

Section Editor

Justin Remais

Section Editor

Reviewer's Responses to Questions

**Key Review Criteria Required for Acceptance?**

**Methods**

-Are the objectives of the study clearly articulated with a clear testable hypothesis stated?

-Is the study design appropriate to address the stated objectives?

-Is the population clearly described and appropriate for the hypothesis being tested?

-Is the sample size sufficient to ensure adequate power to address the hypothesis being tested?

-Were correct statistical analysis used to support conclusions?

-Are there concerns about ethical or regulatory requirements being met?

Reviewer #1: • The authors stated the objective of the study. However, the writers have not spelt out the justification of the study. I suggest the writers give a solid and convincing reasons for this study of which findings has already been established in literature.

• Authors have obtained permission from the appropriate ethic regulatory bodies for the study. Please let the ‘ethics approval and consent to participate’ statement appear at the methods section of the manuscript and not to the end. Also, kindly provide the ethical clearance reference number of the study in the ethic declaration statement. 

• The authors clearly described the appropriate population suitable for the study. However, the description of cases and control, inclusion and exclusion criteria are not clear and should be stated with clarity. 

• The variable; ‘Family history’. Line 153 is not clear. Please kindly clarify this.

• The sample size was determined from sample size calculation and is sufficient to ensure adequate power to address the hypothesis being tested.

Reviewer #2: The methodology described in the manuscript is appropriate for the clearly stated objective. Ethical requirements were fulfilled for for the study.

**Results**

-Does the analysis presented match the analysis plan?

-Are the results clearly and completely presented?

-Are the figures (Tables, Images) of sufficient quality for clarity?

Reviewer #1: • The results are not completely presented. I kindly suggest the following:

- Table headings should be bolded

- Table 1 and 2 are overlapping. I suggest that if a table cannot fit into one page, the authors should break that into sections. Example; Table 1, Table 1 (continued) maintaining the title of the columns under each table sections.

- Table 1: The categories under ‘education is not clear’. I suggest the writers put the categories as: Cannot read and write, Can read and write. Primary, Higher.

- Please in which currency is the characteristic ‘Income’ categorized? I suggest the authors provide this information in parenthesis beside ‘Income’. Example: Income (USD) 

- Table 2: Please the statement in Line 213-214 cannot be found in the referenced table (Table 2)

- Please what is the difference between the result mentioned in Line 210-211 and Line 214-216? Please clarify.

- The characteristic ‘Washing feet per week’, is it without soap? If so, please correct it.

- I think the characteristic ‘Years with shoe wearing should be ‘Number of years of shoe wearing’

- Please the categories under the characteristic ‘Year with shoe wearing’ is not clear. The arrangement of categories is not chronological. What is the difference between ' More than half year and Less than a year? Please correct this statement and let the categories be chronologically presented.

- The characteristic ‘Days with shoe from the week’ not clear. I suggest it should be presented as ‘Number of days of shoe wearing per week’

- Line 235-236: The statistic; 2.05 mentioned in the statement ‘The odds of having podoconiosis was 2.05 times higher….’is not found in the referenced Table 3. Please correct this.

- Table 3: Please provide a column containing information on the P- values

Reviewer #2: The results are clearly presented.

**Conclusions**

-Are the conclusions supported by the data presented?

-Are the limitations of analysis clearly described?

-Do the authors discuss how these data can be helpful to advance our understanding of the topic under study?

-Is public health relevance addressed?

Reviewer #1: • The authors provided conclusions which are supported by the data presented in the manuscript.

• The limitation of the study is mentioned.

• The benefit of the findings to the advancement of knowledge on the topic under study have been discussed and public health relevance has also been touched on.

Reviewer #2: The conclusions are supported by the data presented and study limitation was also noted.

**Editorial and Data Presentation Modifications?**

Reviewer #1: I suggest the authors take into consideration the recommendations made in the results section to make them look more presentable.

Reviewer #2: (No Response)

**Summary and General Comments**

Reviewer #1: • At present, the manuscript needs some further work:

1. The manuscript is poorly written with typographical and grammatical errors making it difficult to read and understand. I therefore highly recommend language editing

2. I presume the word ‘Determinantes’ in the title is supposed to be an English word ‘Determinants’. Please correct.

3. The font type and size of the main write up is different from the one used for the reference lists. I suggest authors edit the manuscript to have a uniform font type and size.

4. The ‘Author’s summary’ as required by the journal is missing in the manuscript. I suggest the authors provide this important information after the ‘Abstract’ statement

5. The in-text citation is incorrect. The journal requires a square bracket not parenthesis

6. Most of the bibliographies are incorrect. For example, some are incomplete, some in capital letters etc. I recommend that the authors edit the bibliographies to suite the journal’s style.

Reviewer #2: The study provides important information on determinant of a neglected NTD and adds to the limited body of knowledge on this disease. 

However, this reviewer will recommend that authors should consider employing the service of an English language editor. The grammar usage in the manuscript is currently not at an acceptable level for publication.

PLOS authors have the option to publish the peer review history of their article (what does this mean?). If published, this will include your full peer review and any attached files.

Reviewer #1: No

Reviewer #2: No
---

## [Editor Report · Decision Letter 1]

8 Aug 2023

Dear Mr. Wassie,

Thank you very much for submitting your manuscript "Determinants of Podoconiosis among Residents of Machakle District East Gojjam Zone Amhara Region Ethiopia" for consideration at PLOS Neglected Tropical Diseases. As with all papers reviewed by the journal, your manuscript was reviewed by members of the editorial board and by several independent reviewers. In light of the reviews (below this email), we would like to invite the resubmission of a significantly-revised version that takes into account the reviewers' comments. 

We cannot make any decision about publication until we have seen the revised manuscript and your response to the reviewers' comments. Your revised manuscript is also likely to be sent to reviewers for further evaluation.

Sincerely,

Victor S. Santos, Ph.D

Section Editor

Justin Remais

Section Editor
---

## [Editor Report · Decision Letter 2]

27 Sep 2023

Dear Mr. Wassie,

We are pleased to inform you that your manuscript 'Determinants of Podoconiosis among Residents of Machakle District East Gojjam Zone Amhara Region Ethiopia' has been provisionally accepted for publication in PLOS Neglected Tropical Diseases.

Best regards,

Victor S. Santos, Ph.D

Section Editor

Justin Remais

Section Editor

---

## [Editor Report · Acceptance letter]

1 Oct 2023

Dear Mr. Wassie,

We are delighted to inform you that your manuscript, "Determinants of Podoconiosis among Residents of Machakle District East Gojjam Zone Amhara Region Ethiopia," has been formally accepted for publication in PLOS Neglected Tropical Diseases.

Best regards,

Shaden Kamhawi

co-Editor-in-Chief

Paul Brindley

co-Editor-in-Chief
